# Prediction model of land surface settlement deformation based on improved LSTM method: CEEMDAN-ICA-AM-LSTM (CIAL) prediction model

**Shengchao Zhu**[1], **Yongjun Qin**[1,2]*, **Xin Meng**[1], **Liangfu Xie**[1,2], **Yongkang Zhang**[3], **Yangchun Yuan**[1]

1 College of Civil Engineering and Architecture, Xinjiang University, Urumqi, Xinjiang, China, 2 Xinjiang Civil Engineering Technology Research Center, Urumqi, Xinjiang, China, 3 CCFEB Civil Engineering Co., Ltd., Central South University, Changsha, Hunan, China

* qyjjg@xju.edu.cn

**Data Availability Statement:** All relevant data are within the manuscript and its Supporting Information files.

## Abstract

The uneven settlement of the surrounding ground surface caused by subway construction is not only complicated but also liable to cause casualties and property damage, so a timely understanding of the ground settlement deformation in the subway excavation and its prediction in real time is of practical significance. Due to the complex nonlinear relationship between subway settlement deformation and numerous influencing factors, as well as the existence of a time lag effect and the influence of various factors in the process, the prediction performance and accuracy of traditional prediction methods can no longer meet industry demands. Therefore, this paper proposes a surface settlement deformation prediction model by combining noise reduction and attention mechanism (AM) with the long short-term memory (LSTM). The complete ensemble empirical mode decomposition with adaptive noise (CEEMDAN) and independent component analysis (ICA) methods are used to denoise the input original data and then combined with AM and LSTM for prediction to obtain the CEEMDAN-ICA-AM-LSTM (CIAL) prediction model. Taking the settlement monitoring data of the construction site of Urumqi Rail Transit Line 1 as an example for analysis reveals that the model in this paper has better effectiveness and applicability in the prediction of surface settlement deformation than multiple prediction models. The RMSE, MAE, and MAPE values of the CIAL model are 0.041, 0.033 and 0.384%; $R^2$ is the largest; the prediction effect is the best; the prediction accuracy is the highest; and its reliability is good. The new method is effective for monitoring the safety of surface settlement deformation.

## 1 Introduction

With the growth in population and increasing urbanization, the development and use of urban underground space are rapidly increasing. Due to its geological complexity, lengthy construction cycle, variety of influencing factors, and high professionalism requirements, this

**Funding:** This work was supported by the Natural Science Foundation of Xinjiang Autonomous Region of China.[grant number 2021D01C073]. The funders had no role in study design, data collection and analysis, decision to publish, or preparation of the manuscript.

**Competing interests:** The authors have declared that no competing interests exist.

type of construction involves high risks. Therefore, the accurate prediction and effective control of surface settlement is crucial.

Scholars at home and abroad have achieved remarkable results in research on ground settlement deformation caused by subway excavation. Traditionally, theoretical analysis and calculation, empirical formula fitting, numerical simulation, and physical model experimentation have been employed. In recent years, artificial intelligence algorithms, including BP neural network [1], recurrent neural network (RNN) [2], support vector machines [3,4], grey prediction model [5], and random forest method [6], have been widely used in various fields due to their impressive speed and accuracy. Backpropagation neural network (BPNN) [7], as a representative of the prediction algorithm of surface subsidence, has become mainstream.

The current improvement methods involve two primary considerations. First, geotechnical engineering has significant uncertainty and fuzziness [8]. The relationship between surface settlement deformation and numerous influencing factors is long-term complex nonlinear, which means it cannot be described by simple functions. Second, the construction process data may be impacted by external factors and human involvement. Nonetheless, several methods do not eliminate such interference, thus requiring pre-analytical noise reduction processing of the resulting data. Therefore, a single algorithm is no longer sufficient to meet requirements [9], hence the emergence of combined algorithms.

Moghaddasi et al. [10] utilized an Independent Component Analysis- Artificial Neural Network(ICA-ANN) model to predict maximum surface settlement to enhance reliability. The ICA was employed to optimize the ANN and determining the most desirable values of weights and biases of the neural network layers for more accurate MSS prediction and to avoid trapping in local optimum. Han et al. [11] established the Simulated Annealing- Regularized Extreme Learning Machines(SA-RELM) model, which combines the sinusoidal algorithm and the regularized limit learning machine, to enhance the precision of multifactor prediction. However, these two methods have the disadvantage of few input parameters. Xiao et al. [12] established the AdaBoost gate recurrent unit(GRU) prediction model to improve fitting ability, but they did not consider the impact of human interference. Kim et al. [13] utilized extreme gradient boosting(XGB) integration to learn the super-parameters of the ML algorithm, which achieved optimal prediction performance, strong search ability, easy realization, and superior prediction results. Many scholars have attempted to optimize LSTM models using various methods. Alotaibi et al. [14] utilized convolutional neural networks(CNN) to optimize LSTM models to improve the low learning rate problem. Wang et al.[15] developed a dual contouring—long short-term memory(DC-LSTM) model that can achieve the multistep prediction of trends in time series. Li et al. [16] utilized an Adam-optimized LSTM network to effectively improve prediction accuracy and training speed. Qian Jiangu et al. [17] proposed a wavelet-optimized long short-term memory–auto regressive moving average(LSTM-ARMA) model for ultradeep foundation pit ground settlement analysis, which effectively considered the noise factor and predicted the trend and noise terms separately to reduce the prediction error. However, the influence of the input factor weights was ignored. Khataei et al. [18] utilized the spotted hyena optimizer (SHO) algorithm to optimize the LSTM network for multilabel text classification, and the accuracy rate was significantly higher than that of other models. Yang et al. [19] established an LSTM model that utilizes an attention mechanism to predict the deformation of concrete dams, which effectively considered the influence of significant factors on deformation and temporal variation and improved the prediction accuracy.

The following two main problems have not been addressed in the existing research. First, subsidence monitoring data contain noise pollution because of various factors, such as instrument error, manual error, and the surrounding environment. Second, the prediction process cannot simultaneously consider the variation in the time series and the influence of the characteristics of the input data and the training speed.

To overcome the above problems and obtain subway settlement prediction results faster and more accurately, this paper takes Urumqi Rail Transit Line 1 as an example and combines the field settlement monitoring data. Its main contents are as follows.

1. A subway settlement deformation noise reduction method that fully integrates the adaptive noise of empirical mode decomposition and independent component analysis (ICA) is proposed. The complete ensemble empirical mode decomposition with adaptive noise (CEEMDAN) algorithm is implemented as a dimension raising tool to decompose multidimensional signals, the ICA algorithm functions as a dimension reducing tool to separate the noise reduction signals, and a combined algorithm is utilized to smooth the error in the monitoring process and reduce the impact of noise on the prediction.

2. After a series of developmental steps, the attention mechanism can confirm the allocation weight of input factors through the relationship between input and output. This paper combines an attention mechanism with an LSTM network to construct the AM-LSTM surface settlement deformation prediction model and adds an Adam optimization algorithm with high calculation efficiency to improve prediction speed and accuracy. It can not only consider the influence of long-term dependence in prediction but also capture the influence characteristics during prediction in time while assigning appropriate weights. Compared to several other models, this model has the highest prediction accuracy and the best effect, and thus, it can serve as a reference for the prevention and control of ground settlement deformation caused by subway excavation.

## 2 Theoretical basis

### 2.1 CEEMDAN algorithm

To solve the problem of modal aliasing and end effect caused by the EMD algorithm signal decomposition and the residual white noise problem of EEMD algorithm and CEEMD decomposition, Torres [20] proposed CEEMDAN. Colominas et al. [21] compared EEMD with CEEMDAN and found that only the latter recovered completeness. Zhao et al. [22] found that CEEMDAN both overcame the modal aliasing problem caused by EMD and reduced the residual noise problem caused by EEMD.

The CEEMDAN method differs from other algorithms in that white noise is added to the original signal, but adaptive noise is added to the *IMF* component several times after decomposition using the EEMD algorithm. The first-order component *IMF* of the decomposition is obtained as the overall average value of the first eigenmode component. The same white noise is then added to the residual value to find the first eigenmode component, and all eigenmode components are found iteratively. The original signal is $X(t)$, the Gaussian white noise is $N_f(t)$, and the number of times white noise is added is $f$.

The specific decomposition steps of the CEEMDAN algorithm are as follows.

1. Repeatedly add adaptive noise $N(t)$ to the original signal $X(t)$ to obtain the first noisy signal $X'(t) = X(t) + N_f(t)$, and then conduct EMD decomposition to obtain the first order eigenmode component $IMF'_1$;

2. The overall average value of $IMF'_1$, which is the first-order eigenmode component of the $X(t)$ decomposition $IMF_1$ is obtained, after which the remainder $R_1(t)$ is obtained.

$$IMF_1 = \frac{1}{f} \sum IMF_1{}',$$

(1)

$$R_1(t) = X(t) - IMF_1.$$

(2)

3. A new decomposed signal $X_1(t)$ is constructed, and noise is added to obtain $X_1'(t)$. The

second order intrinsic mode component $IMF'_2$ is then obtained via EMD decomposition.

$$X_1(t) = R_1(t), \tag{3}$$

$$X_1'(t) = X_1(t) + N_f(t). \tag{4}$$

4. The overall average of $IMF'_2$, which is the second-order eigenmode component $IMF_2$ of the $X(t)$ decomposition, and the remaining term $R_2(t)$ can be obtained.

$$IMF_2 = \frac{1}{f} \sum IMF_2', \tag{5}$$

$$R_2(t) = X_1(t) - IMF_2. \tag{6}$$

5. Repeat steps (1) and (2) until the program terminates to obtain all the eigenmodal components $IMF_i$.

## 2.2 ICA algorithm

ICA is a method of separating independent source signals by applying statistical principles to the original observed signals. It plays a prominent role in blind source separation [23,24], feature recognition [25], and signal separation [26]. The source signal $S(t) = \{S_1(t), S_2(t), \ldots, S_n(t)\}$ is estimated from the known mixed signal $X(t) = \{X_1(t), X_2(t), \ldots, X_n(t)\}$, and a certain linear relationship between $X(t)$ and $S(t)$ exists, which can be expressed as

$$X(t) = AS(t). \tag{7}$$

The principle underlying the ICA algorithm is that for a nonzero mean independent source signal $S(t)$, the actual observed signal $X(t)$ is obtained from $X(t) = A\,S(t)$ after data preprocessing, and the unmixing matrix $A$ is obtained, as detailed in the process shown in **Fig 1**.

Among the numerous ICA algorithms, the Fast-ICA algorithm, which is based on negative entropy maximization, is a popular choice for signal processing owing to its computational simplicity, fast and robust convergence, and good separation effect. The implementation steps of the Fast-ICA algorithm are as follows:

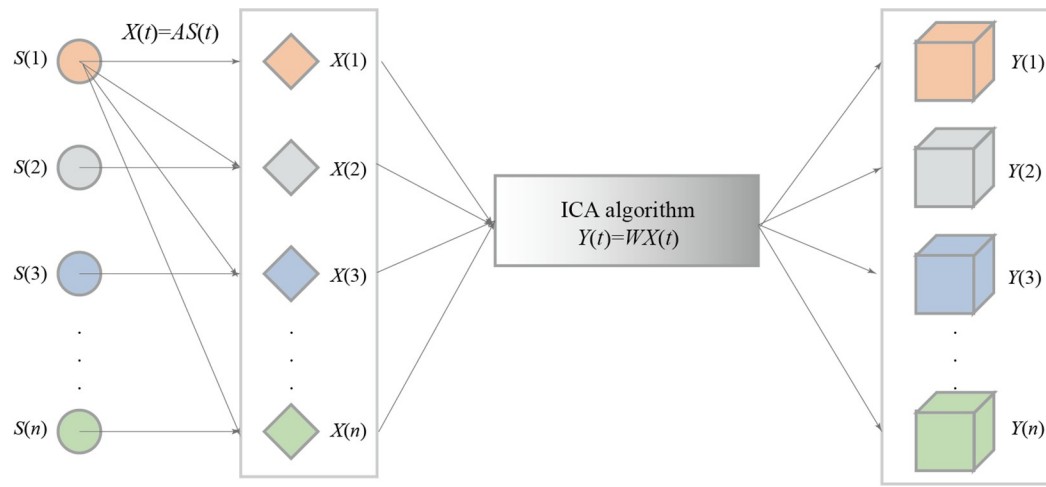

**Fig 1. The model of ICA algorithm.**

1. Transform the original observed signal into a matrix $X(t)$ with n rows and m columns;

2. Decentralize the $X(t)$ data;

$$\bar{X}_j = X_j - \sum_{i=1}^{m} X_j^{(i)}, j = 1, 2, 3, \ldots, n. \tag{8}$$

3. Whiten the data after centralization;

① First, the covariance matrix $C_X = \frac{1}{m} X \cdot X^T$ of the solved data $X(t)$.

② Perform feature solving $C_X V = V\Lambda$.

where $V = [v_1, v_2, \ldots, v_n]$ is the column matrix of eigenvectors, and $\Lambda = \text{diag}(\lambda_1, \lambda_2, \ldots, \lambda_n)$ is the diagonal matrix consisting of eigenvalues of the covariance matrix $C_X$.

③ The final result after whitening of the original observed data signal is obtained as follows.

$$X_{w(n\times m)} = \Lambda^{\left(-\frac{1}{2}\right)}_{(n\times n)} \cdot V^T_{(n\times n)} \cdot X_{(n\times m)} \tag{9}$$

4. Set the value of the magnitude of the parameter learning rate $\alpha$;

5. Solve for the unmixing matrix $W$ at moment $i$;

6. Solve the estimated independent component signal $S_{n\times 1}^{(i)} = W_{n\times n} \cdot X_{n\times 1}$ at moment $i$;

7. Repeat steps (3)–(6) to obtain the independent component signals $S_{n\times m} = [s^{(1)}, s^{(2)}, \ldots, s^{(m)}]$ for all moments.

## 2.3 CEEMDAN-ICA noise reduction model

The CEEMDAN algorithm has its advantages and is suitable for feature extraction and signal noise reduction during time domain analysis of nonsmooth, nonlinear data [27–29] but still suffers from small amounts of noise, low effective signal loss, and slow iteration speed [30,31]. The ICA algorithm can improve the accuracy and speed of signal separation, and CEEMDAN-ICA joint noise reduction combines the advantages of both CEEMDAN and ICA to build on their strengths and avoid their weaknesses. It has been widely applied, including in electromagnetic signals [32], EEG signals [33,34], and LIDAR systems [35] but has rarely been used in surface settlement deformation noise reduction.

Principally, the CEEMDAN algorithm is used to decompose the initial observation signal to obtain a series of intrinsic mode components. According to the correlation criterion, the *IMF* components with large noise are judged, and then the noise *NIMF* components obtained from the sum form a dual channel input signal with the original observation signal. Fast-ICA algorithm is used to separate and reconstruct the signal, and the effective signal after noise reduction can then finally be obtained. The steps are as follows, and the process is detailed in **Fig 2**.

1. The CEEMDAN decomposition of the initial observed signal $X(t)$ is performed to obtain the eigenmodal components $IMF_1$, $IMF_2$,. . . . . .,., and $IMF_n$ of the original observed signal.

2. The correlation between the original observation signal $X(t)$ and each *IMF* is calculated, and the Spearman correlation coefficient is adopted. The principle is shown in Formula (1), and its size is [–1,1].

$$\rho(X(t), IMF) = \frac{\sum_{i=1}^{n}(X(t) - \bar{X(t)})(IMF_n(t) - I\bar{M}F_n)}{\sqrt{\sum_{i=1}^{n}(X(t) - \bar{X(t)})^2 \sum_{i=1}^{n}(IMF_n(t) - I\bar{M}F_n)^2}} \tag{10}$$

where n is the sample size, $\rho$ is the correlation coefficient, $X(t)$ is the data of the initial observed signal, $IMF_n(t)$ is the *IMF* component, and $I\bar{M}F_n$ is the mean of the *IMF* component.

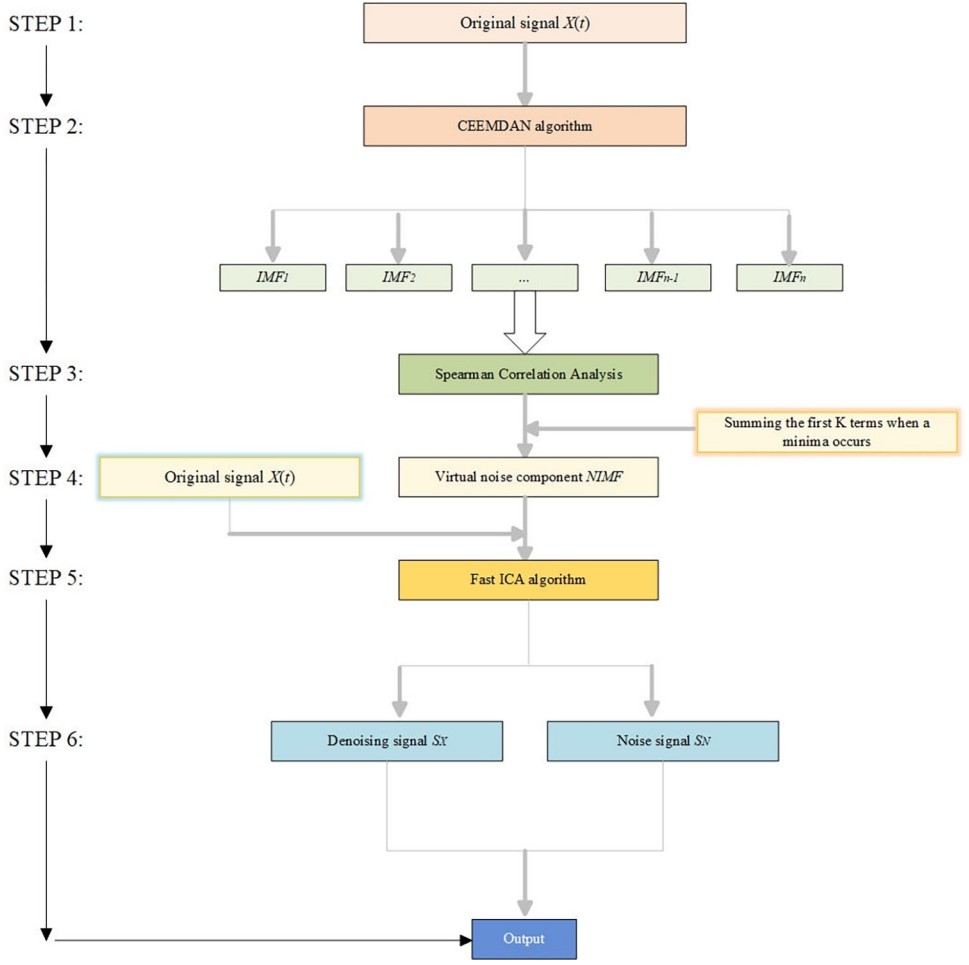

**Fig 2. CEEMDAN-ICA noise reduction principle.**

3. The virtual noise component *NIMF* signal is derived based on the fact that the noise component is the sum of the first *K IMF* components when the correlation coefficient has a minimal value in the previous *K* terms [36].

4. The *NIMF* signal and original observation signal $X(t)$ are used as inputs of the Fast-ICA algorithm for signal noise separation, and source signals $S_1$ and $S_2$ are output.

5. The application based on the MDP criterion [37,38] can revise and eliminate the problems of incomplete recovery of the source signal amplitude, uncertainty of the source signal order, and uncertainty of the positive and negative sign of the source signal resulting from the independent components estimated by the ICA algorithm to obtain an effective noise reduction signal $S_X$ and noise signal $S_N$.

## 2.4 LSTM neural network

Gers et al. [39] conducted a review that indicated that the standard LSTM algorithm outperforms other RNN algorithms on illustrative benchmark problems. The LSTM network structure is an improved RNN, which introduces the idea of gating. It remembers information through forgetting, memory, and output gates and is used to add, remove, retain, and transmit the state and information of the incoming and outgoing units. The four neural network layers

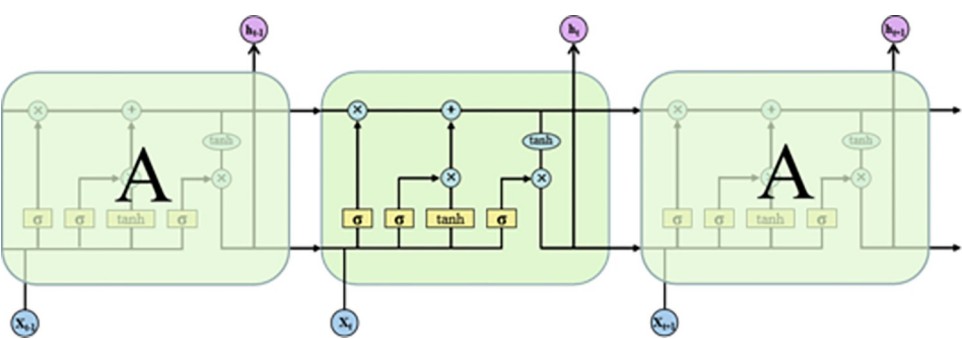

**Fig 3. LSTM unit structure [40].**

interact in a special way, which can effectively avoid the long-term dependence of an RNN. The LSTM model first filters the subway settlement deformation input variables from the memory cell through the forgetting gate, and the input gate identifies important information. Finally, the output gate outputs the subway settlement deformation data and transmits it to the next LSTM structural unit. The modeling process inside each LSTM structural unit is as follows, and the process is detailed in **Fig 3**.

1. Forgetting gate: receives the output information $h_{t-1}$ of the previous moment and the input information $X(t)$ of the current moment and performs forgetting screening by multiplying them together to retain the desired content.

$$f_t = \sigma(W_f \cdot [h_{t-1}, x_t] + b_f). \tag{11}$$

2. Input gate: determines that information retained by the forgetting gate that needs to be updated and stored in the immediate cell state $\tilde{C}_t$ and filters the information, updates the cell state using the sigmoid activation function, and maps the output information to the [0,1] interval.

$$i_t = \sigma(W_i \cdot [h_{t-1}, x_t] + b_i), \tag{12}$$

$$\tilde{C}_t = \tanh(W_C \cdot [h_{t-1}, x_t] + b_C), \tag{13}$$

$$C_t = f_t * C_{t-1} + i_t * \tilde{C}_t. \tag{14}$$

$$P_t = \sigma(W_o \cdot [h_{t-1}, x_t] + b_o) \tag{15}$$

$$h_t = P_t * \tanh(C_t) \tag{16}$$

3. Output gate: determines $h_{t-1}$ and $X(t)$ information that will be output. The cell state $C_t$ is compressed to the (-1, 1) interval by the tanh activation function, and the hidden state $h_t$ at the current moment is obtained as the output through the output gate.

where $\cdot$ represents the matrix product and $*$ denotes the Hadamard product, that is, the product of elements.

## 2.5 Attention mechanism

The attention mechanism can assign different weights from a large amount of information depending on the characteristics of each layer of the network to highlight the more important information and constrain the less important information [41]. The specific calculation

process of the attention mechanism can be divided into three steps: first, the correlation or similarity between query and key is calculated according to their vector dot product; second, the Softmax function is introduced to normalize the weights obtained in the first stage, and the internal mechanism of the Softmax function is used to highlight the key information; finally, the attention value can be obtained by weighting the sum of the weight and its corresponding value.

$$Attention(q, k, v) = Softmax\left(\frac{qk^{T}}{\sqrt{d^{f}}}\right).$$

(17)

Here, $q$ is the abbreviation of query, which is a query vector and is used to query; $k$ is the abbreviation of key, which is a key vector and serves as an index; $v$ is the abbreviation of value, which is the value vector, and its role is the main content; and $d^{f}$ represents the characteristic dimension of the input information, which is used to enhance the stability of the Softmax function during calculation. The calculation process of $q$, $k$, $v$ is as follows:

$$q = W^{q}X,$$

(18)

$$k = W^{k}X,$$

(19)

$$v = W^{v}X.$$

(20)

The monitoring data $X$ are multiplied by $W^{q}$, $W^{k}$, and $W^{v}$, respectively, and linearly transformed to obtain $q$, $k$, $v$. $W^{q}$, $W^{k}$, and $W^{v}$ need to be customized at the time of calculation and continuously debugged according to the results. Different weight assignments will give different results, which in turn affect the prediction effect, so the best combination should be selected.

## 2.6 AM-LSTM prediction model

Zheng et al. [5] illustrated the underlying rationale behind the excellent performance of the attention mechanism in learning long-term dependencies by studying the memory properties of LSTM networks with the attention mechanism, and the results revealed that the decline due to the attention mechanism is significantly slower than that of the LSTM. Moreover, the LSTM model containing the attention mechanism not only performs better in time series prediction but also maintains long-time memory while significantly lowering the training time. The validation was performed successively by Chen and Zhang [31].

The structure of an AM-LSTM model in this paper can be divided into five parts: input, LSTM, attention, fully connected, and output layers. The role of the LSTM layer is to sense the state, memorize the information, and perform feature learning; the attention layer redistributes different features and highlights the key information, and the fully connected layer performs local feature integration to achieve the final prediction.

Input layer: The signal $S_{X}$ after noise reduction is normalized to obtain $I(t)$ and input to the model. The normalization process aims to speed up the convergence and improve the accuracy of the prediction model. The data are scaled and linearly transformed through min-max normalization so that the result is between 0 and 1. The principle formula is as follows.

$$I(t) = \frac{S_{X} - S_{min}}{S_{max} - S_{min}}$$

(21)

where $I(t)$ is the data after data normalization, $S_{max}$ is the maximum value of the data $S_{X}$ after noise reduction, and $S_{min}$ is the minimum value of the data $S_{X}$ after noise reduction.

LSTM layer: The LSTM layer is used to learn the input sequence $I(t)$ to get the output sequence $L(t)$ of the LSTM at moment $t$.

Attention layer: The input vector $L(t)$ of the LSTM layer is used as the input of the attention layer, and the feature vectors are weighted and summed to highlight the key information to obtain the output sequence $\alpha t$ of attention at moment $t$.

Fully connected layer: The output result $\alpha t$ of the attention layer is used as the input of the fully connected layer, the normalized data result of each input data is introduced to train the model, and the normalized prediction value $Y(t)$ of the model output is obtained.

Output layer: The output prediction value $Y(t)$ of the fully connected layer is denormalized to obtain the final prediction result $P(t)$, and the process is detailed in **Fig 4**.

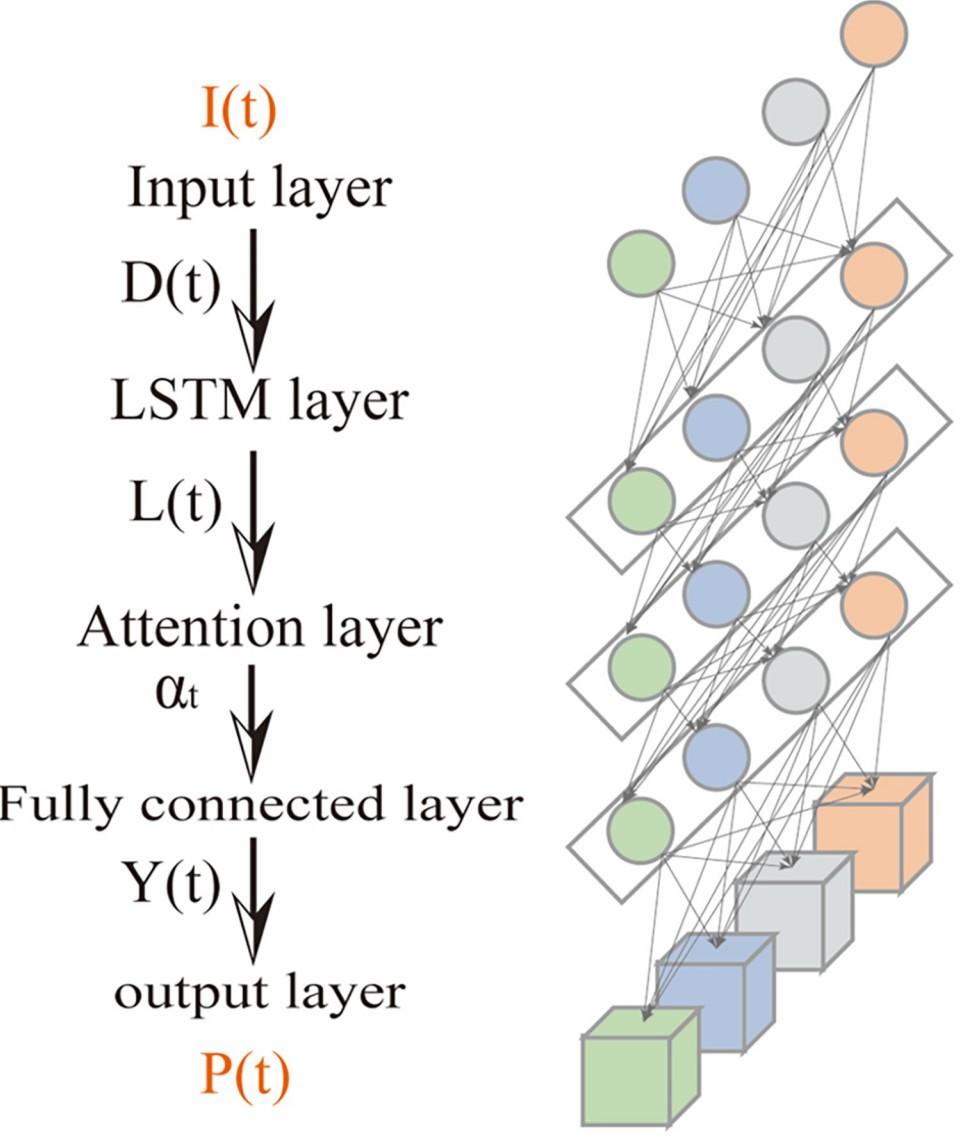

**Fig 4. Attention–LSTM model diagram.**

## 3 Process and analysis

### 3.1 Dataset acquisition

The data in this paper were obtained from the Wangjialiang Station Project of Urumqi Rail Transit Line 1. The station body is an 11 m platform island-type underground excavation station with two underground layers. The standard section adopts a box frame structure with a width of 20.1 m and a height of 15.29 m. During the excavation of the foundation pit, the main strata are the pebble layer, silty clay, mudstone, and sandstone, among others, and bedrock fissure water is present, which makes construction challenging.

The layout of the surface settlement measuring points of Wangjialiang Station is shown in **Fig 5**(A) and 5(B). At present, the settlement of each measuring point has converged, the settlement rate is generally less than ±2.00mm/d, and the cumulative settlement is between -21.00 and 6.00mm. The prediction model in this paper collected the 201 phase settlement monitoring data points of test point DB-37-03 to establish a data set. The ratio of the training set to the test set is 5:1. The first 176 phases of data are used as the training set of the prediction model, whereas the last 26 phases of data are used as the test set of the prediction model to verify the feasibility of the prediction model proposed in this paper. The longitudinal geological profile of the location of test point DB-37-03 is shown in **Fig 5**(C).

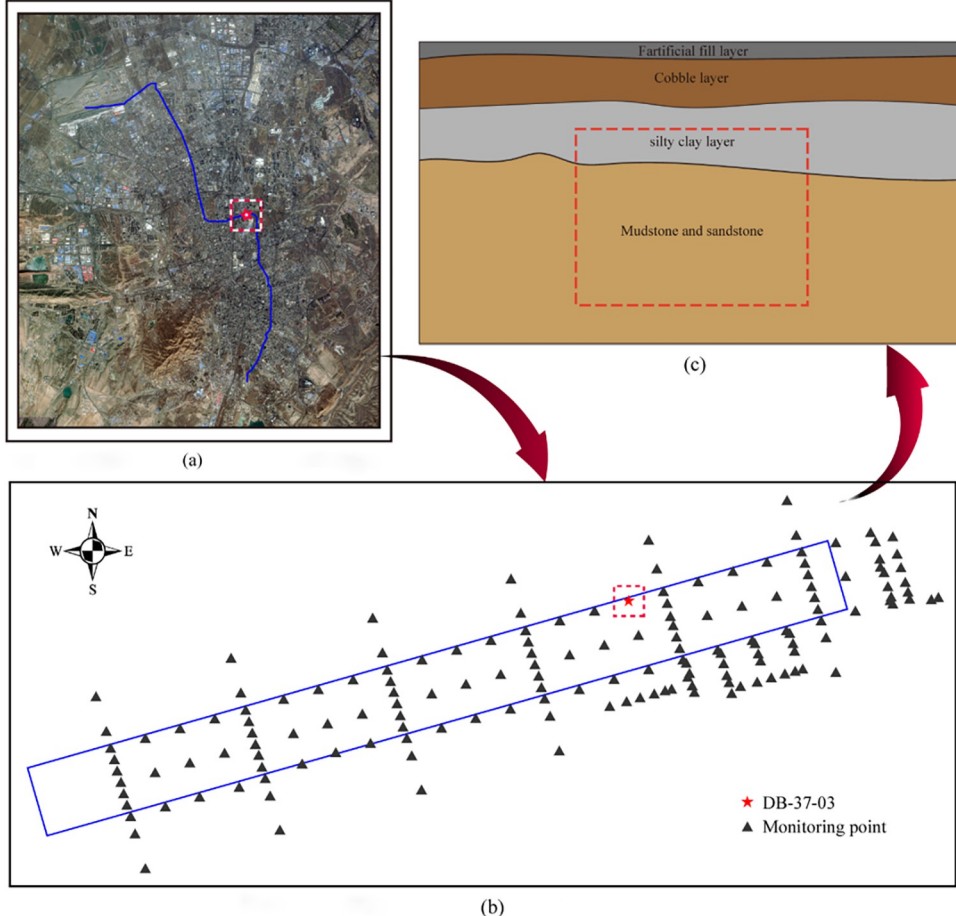

**Fig 5. Wangjialiang station diagram.** (a) Urumqi rail transit line 1 project line plane diagram; (b) Wangjialiang station surface subsidence measuring point layout; (c) Geological profile.

## 3.2 Evaluating indicator

To verify the effectiveness of the algorithm, the correlation coefficient and root mean square error (RMSE) were used to evaluate the effect of noise reduction. RMSE, mean absolute error (MAE), mean absolute percentage error (MAPE), and sample regression determination coefficient ($R^2$) were used as the evaluation indicators of CEEMDAN-ICA-AM-LSTM prediction model in this paper. RMSE is used to measure the deviation between the noise-reduced data and the original data and the deviation between the predicted and observed values, reflecting the accuracy of the prediction. MAE is a measure of the average error in prediction; MAPE reflects the degree of error fluctuation in prediction, which can be used to evaluate the stability of the model; and $R^2$ reflects the degree of fitting of the predicted data to the original measured data. The larger the correlation coefficient, the smaller the RMSE; the better the noise reduction effect; and the smaller the RMSE, MAE, and MAPE, whereas the larger the $R^2$, the better the prediction effect.

$$RMSE = \sqrt{\frac{1}{n}\sum_{t=1}^{n}(P(t) - X(t))^2} \tag{22}$$

$$MAE = \frac{\sum_{t=1}^{n}|P(t) - X(t)|}{n} \tag{23}$$

$$MAPE = \frac{100\%}{n}\sum_{i=1}^{n}\left|\frac{P(t) - X(t)}{X(t)}\right| \tag{24}$$

$$R^2 = 1 - \frac{\sum_{t=1}^{n}(P(t) - X(t))^2}{\sum_{t=1}^{n}(X(t) - \bar{X})^2} \tag{25}$$

where n is the number of data samples, $P(t)$ is the predicted data, $X(t)$ is the measured data, and $\bar{X}$ is the average of the measured data $X(t)$.

## 3.3 Data noise reduction process

The original observation signal of measuring point DB-37-03 collected at Wangjialiang station was decomposed using the CEEMDAN algorithm, which can be decomposed into five eigenmode component $IMF$s ($IMF_1$–$IMF_5$) and one residual component ($IMF_6$), as shown in **Fig 6**.

Then, according to the principle of Spearman's correlation coefficient, the correlation coefficient of each $IMF$ component with the original observed data signal was calculated, as shown in **Fig 7**. The first minimal value point appears in the third eigenmode component $IMF_3$, and the first three $IMF$ components can be judged as noise components reconstructed to obtain the virtual noise component $NIMF$.

Next, the virtual noise component $NIMF$ signal and the original observation data of Wangjialiang station are used as the dual input channels of the Fast-ICA algorithm, and the source signals $S_1$ and $S_2$ are output (**Fig 8**). The correlation coefficient between signal $S_1$ and the original observation signal is 0.9995 according to the Spearman correlation coefficient principle, and the correlation is very high, so the signal can be judged as the noise reduction signal. However, signal $S_1$ has the problem that amplitude and phase cannot be fully recovered, and revision by applying the MDP criterion is necessary to finally obtain the effective noise reduction signal $S_X$ (**Fig 9**) and the noise signal $S_X$ (**Fig 10**).

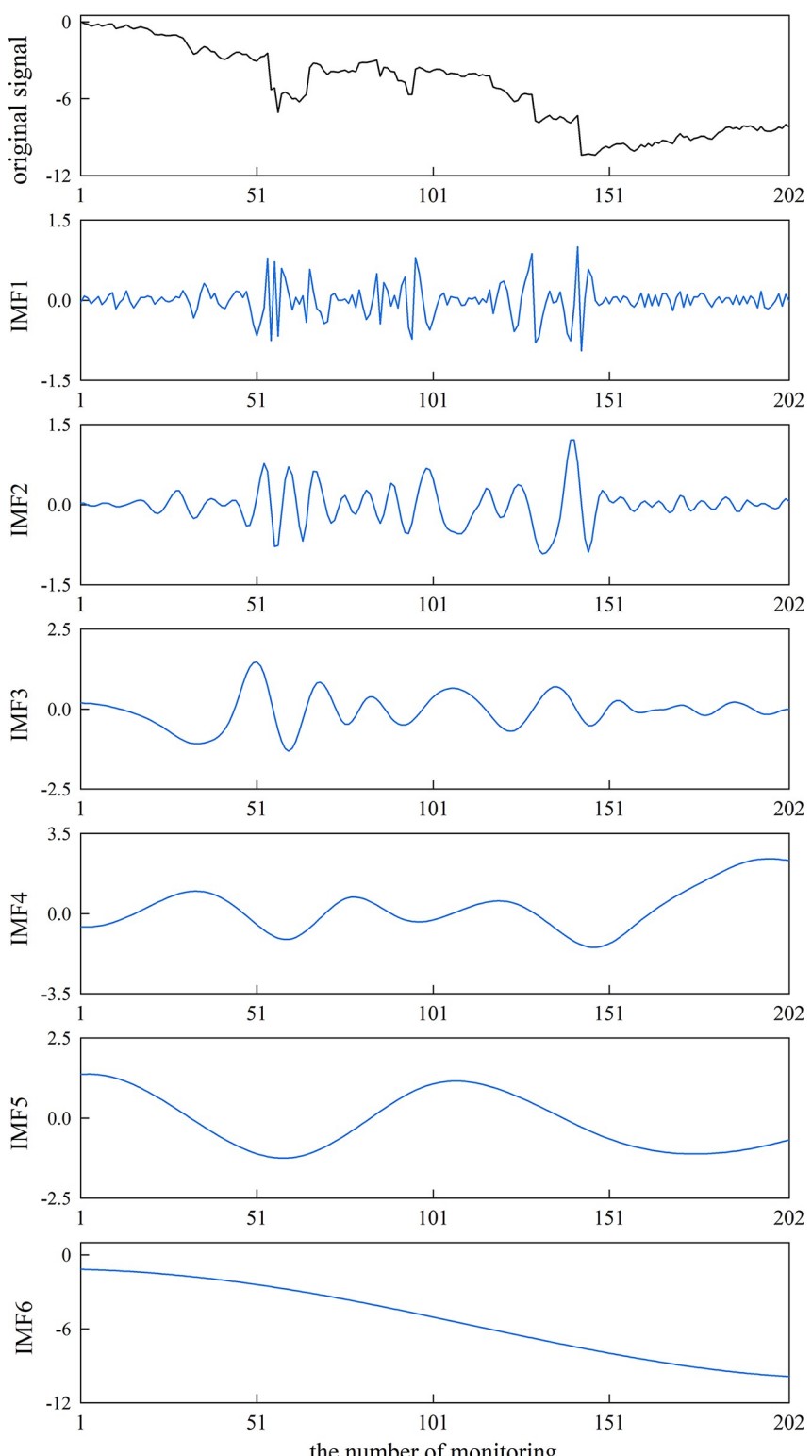

**Fig 6. IMFS components of DB-37-03 signal decomposition in Wangjialiang station.**

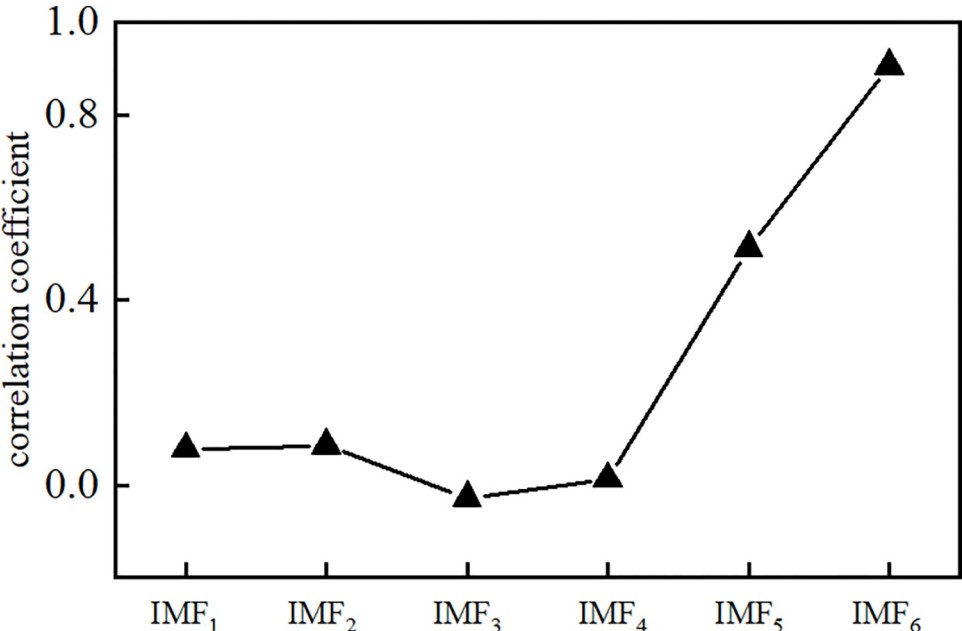

**Fig 7. Correlation coefficient between *IMF_S* components.**

Compared with the original observation data $X(t)$ of measuring point DB-37-03 at Wangjialiang Station, the observation data $S_X$ after noise reduction by CEEMDAN-ICA model filtering have less noise and a smoothed curve trend. Meanwhile, the evaluation indicators are used

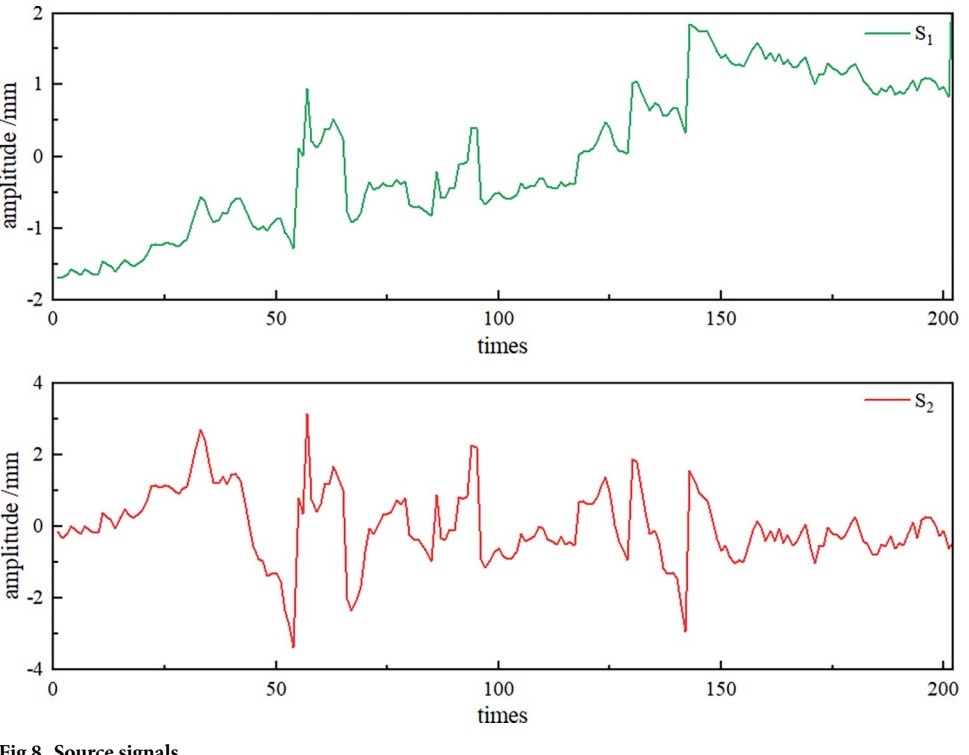

**Fig 8. Source signals.**

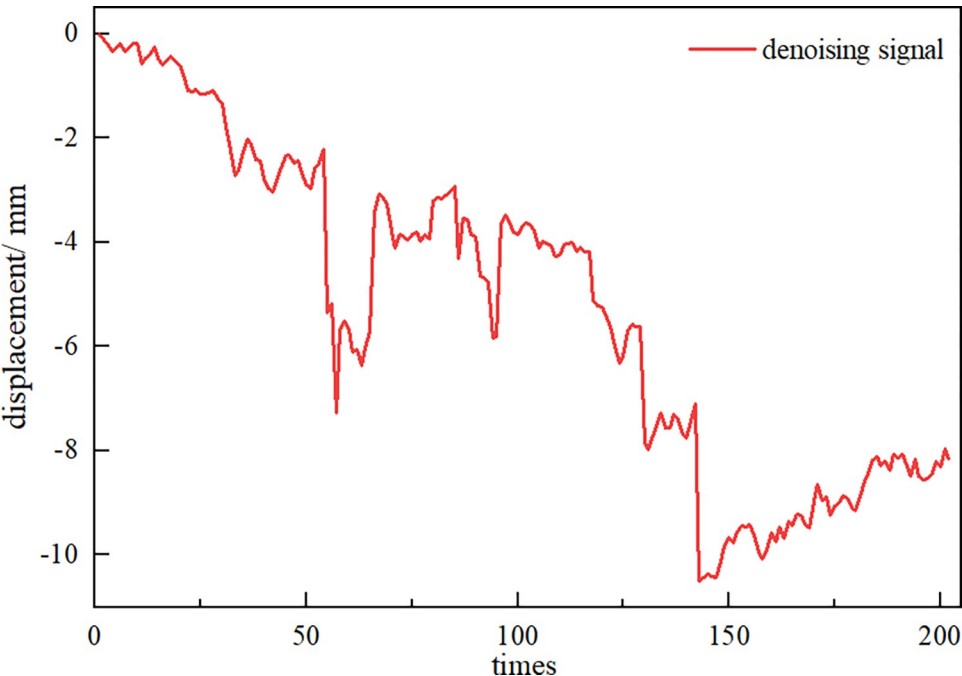

**Fig 9. Denoising signal $S_X$.**

for quantitative comparison. See Table 1 for details. The median of the data after noise reduction is largely unchanged compared to the original observed data, the minimum value becomes larger, and the variance exhibits a significant downward trend, decreasing by 0.139. The results reveal that the CEEMDAN-ICA noise reduction model can reduce the adverse effects of noise

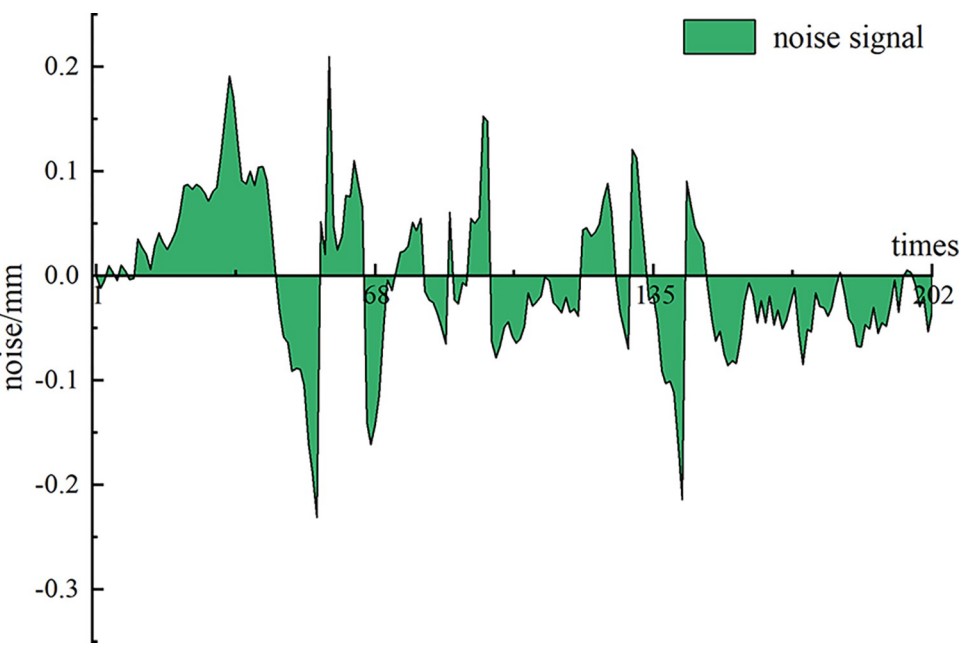

**Fig 10. Noise signal $S_N$.**

**Table 1. Analysis indexes before and after noise reduction.**

| Analysis indicators | X(t)/mm | $S_X$/mm |
|---|---|---|
| Maximum value | 0 | 0 |
| Minimum value | -10.400 | -10.226 |
| Median | -3.735 | -3.738 |
| Square difference | 2.733 | 2.705 |
| Variance | 9.720 | 9.581 |

while ensuring that the overall trend characteristics of the original observed data remain unchanged.

To further illustrate the effectiveness of the method in this paper, the data of several identical stations and the same measurement points in Jiaqi Zhang's paper were selected to compare it with the improved wavelet noise reduction method. Table 2 indicates that the RMSE values of the observed data after the noise reduction of the CEEMDAN–ICA model are prominently lower, indicating that the component contains more features of the original signal, and the noise reduction effect is pronounced, which confirms the effectiveness and feasibility of the CEEMDAN–ICA noise reduction model in the metro deformation monitoring data.

### 3.4 Model training

To increase the validity and feasibility of the prediction models, the prediction effects of LSTM, AM-LSTM (hereinafter referred to as AL), CEEMDAN-ICA-LSTM (hereinafter referred to as CIL), and the CEEMDAN-ICA-AM-LSTM (hereinafter referred to as CIAL) prediction models were analyzed in a cross-sectional comparison using the same data set in this paper. After pre-debugging, the training times of all four models were 700, the training steps were 5, the prediction step was 1, the selected optimizer was the Adam algorithm, and the loss function of the prediction model was the RMSE loss function to facilitate gradient descent and function convergence. The specific process can be seen in **Fig 11**.

After determining the parameters of the LSTM, AL, CIL, and CIAL prediction models, the training pairs were trained iteratively using the training set, and the prediction results were compared and analyzed on the test set; the results are shown in **Fig 12**.

## 4 Result and discussion

### 4.1 Performance comparison

The prediction performance of the four prediction models, LSTM, AL, CIL, and CIAL on the training set of Wangjialiang station is listed in Table 3.

Compared with the LSTM prediction model, the RMSE value of CIAL model decreased from 0.139 to 0.065 (52.98%), the MAE value decreased from 0.111 to 0.061 (45.27%), the $R^2$ value increased from 0.798 to 0.955 (19.67%), and the MAPE value decreased from 1.1319% to 0.729% (44.73%). Compared to the CIL prediction model, the RMSE value of CIAL model

**Table 2. RMSE comparison of noise reduction models at different sites.**

| Site name | Wavelet noise reduction | Method of this article |
|---|---|---|
| Nanhu North Road station | 0.2759 | 0.0612 |
| Wangjialiang station | 0.2785 | 0.1038 |
| Santunbei—Xinjiang University | 0.1161 | 0.0636 |
| Nanhu North Road—Wangjialiang | 0.2603 | 0.2427 |

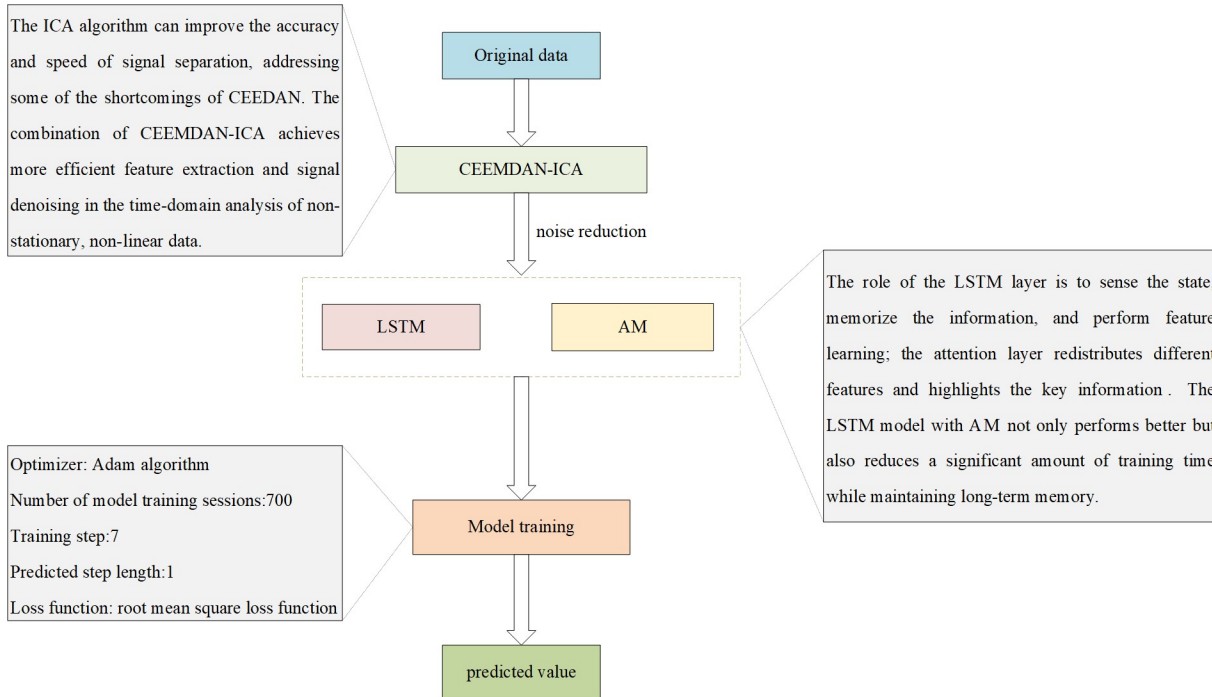

**Fig 11. The training specific process and methods of the model.**

decreased from 0.097 to 0.041 (58.03%), the MAE value decreased from 0.079 to 0.033 (58.30%), the $R^2$ value increased from 0.903 to 0.983 (8.86%), and the MAPE value decreased from 0.938% to 0.384% (59.06% year-on-year decrease). Comparing the two groups of models reveals that the prediction ability and effect of the model with the attention mechanism is

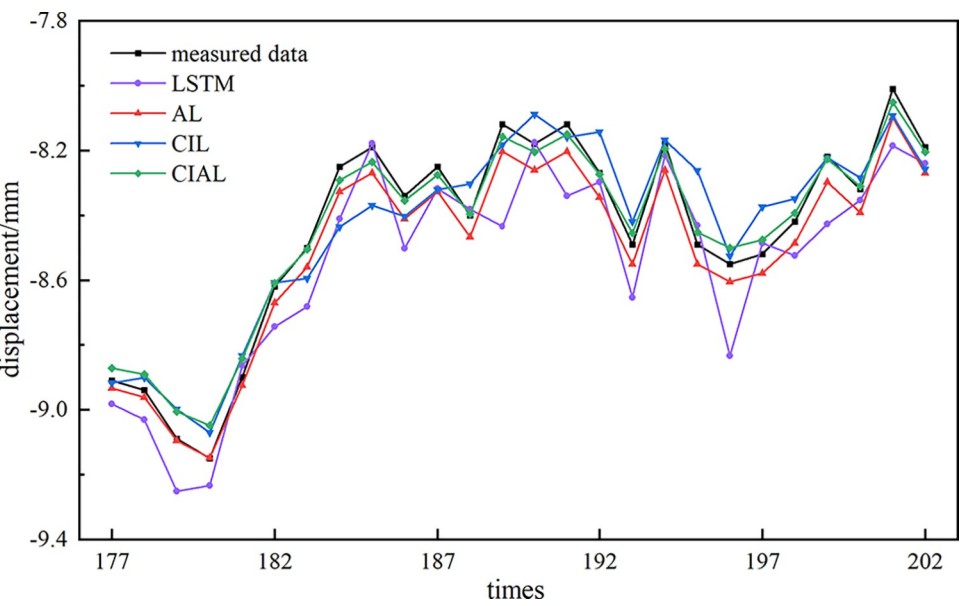

**Fig 12. Model prediction results.**

**Table 3. Four model prediction evaluation indicators.**

| Predictive Models | RMSE (mm) | MAE (mm) | $R^2$ | MAPE (%) |
|---|---|---|---|---|
| LSTM | 0.139 | 0.111 | 0.798 | 1.319 |
| AL | 0.065 | 0.061 | 0.955 | 0.729 |
| CIL | 0.097 | 0.079 | 0.903 | 0.938 |
| CIAL | 0.041 | 0.033 | 0.983 | 0.384 |

significantly better than that of the model without the attention mechanism, indicating that the attention mechanism can effectively improve the prediction accuracy of the model.

Compared to the LSTM prediction model, the CIL prediction model shows a 30.63% decrease in RMSE values from 0.139 to 0.097, a 28.85% decrease in MAE values from 0.111 to 0.079, a 13.10% increase in $R^2$ values from 0.798 to 0.903, and a 13.15% decrease in MAPE values from 1.1319% to 0.983%. Compared to the AL prediction model, the RMSE value decreased by 38.07% from 0.065 to 0.041, the MAE value decreased by 45.80% from 0.061 to 0.033, the $R^2$ value increased by 8.89% from 0.955 to 0.983, and the MAPE value decreased by 47.32% from 0.729% to 0.384%. Comparing the two groups of models reveals that the prediction ability and effect of the filtered noise-reduced model are significantly better than those of the original model. This, combined with the visualization results in **Fig 13**, suggests that the original metro deformation monitoring sequence containing random noise and monitoring anomalies will interfere with the robustness of the prediction model to a certain extent and have some effect on its prediction accuracy. It also indicates that the CEEMDAN-ICA noise reduction algorithm can effectively smooth the subway settlement data with nonlinearity and nonsmoothness, which can effectively improve the prediction accuracy of the model.

Table 3 and **Fig 13** reveal that the performances of the four prediction models in terms of the decision coefficient $R^2$ are largely the same, and CIAL has the best prediction performance. Compared with the other three prediction models, the CIAL prediction model has the smallest residual value; the smallest RMSE, MAE, and MAPE values; and the largest $R^2$. Each index is the best, and the prediction accuracy is optimal.

## 4.2 Robustness testing

Junqi Yu et al. [42] used absolute error comparison analysis to verify that the TSA-RBF-LSTM model has good stability. In this paper, the absolute error AE = |predicted value-observed value| was obtained for the four prediction models and plotted in a box plot. **Fig 14** shows that the CIAL prediction model has the smallest error magnitude range, and the median and mean values are closest to "0" compared to those in the other three prediction models, which indicates that the prediction model can control the absolute error of the prediction results within a small value interval. These results demonstrate that the model is robust and has strong prediction performance.

## 4.3 Evaluation of fit

The distribution of the four models LSTM, AL, CIL, and CIAL in **Fig 15** on the test set showed that all the points were on both sides of the function $y = x$, which is relatively consistent, and the correlation coefficient was more than 0.79. No overfitting or underfitting occurred. However, compared with other prediction models, the CIAL prediction model had the smallest deviation from the true value in the prediction of subway settlement, and it performed best among all networks. This confirms that the model has better prediction accuracy and prediction performance than the other models.

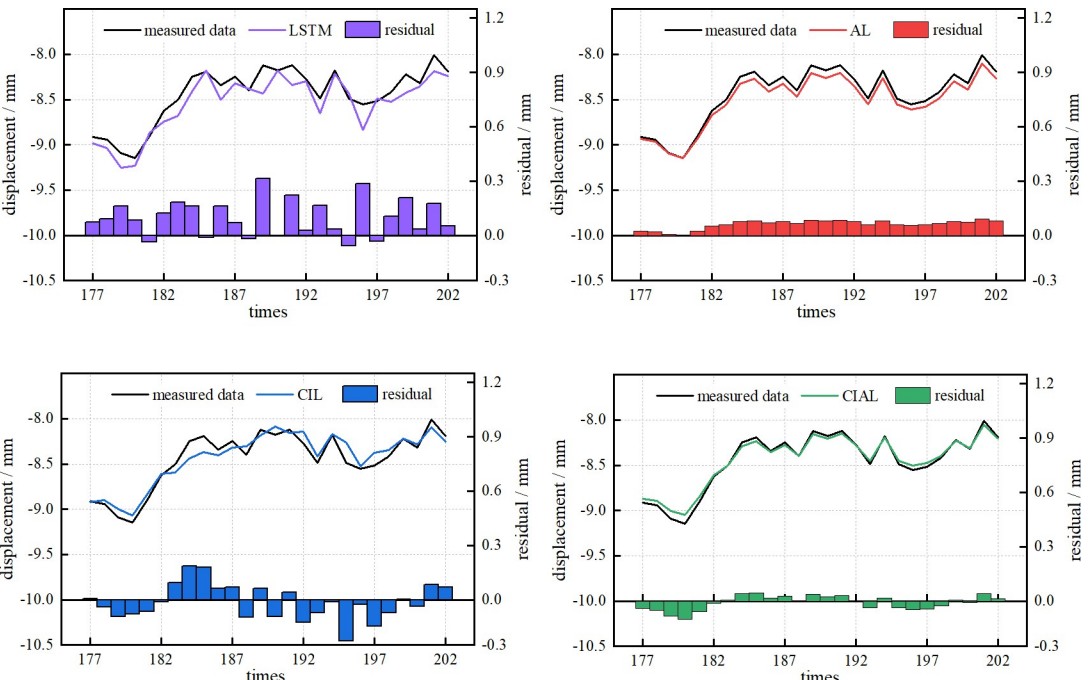

**Fig 13. Comparison of prediction results.** (a) LSTM model prediction results; (b) AL model prediction results; (c) CIL model prediction results; (d) CIAL model prediction results.

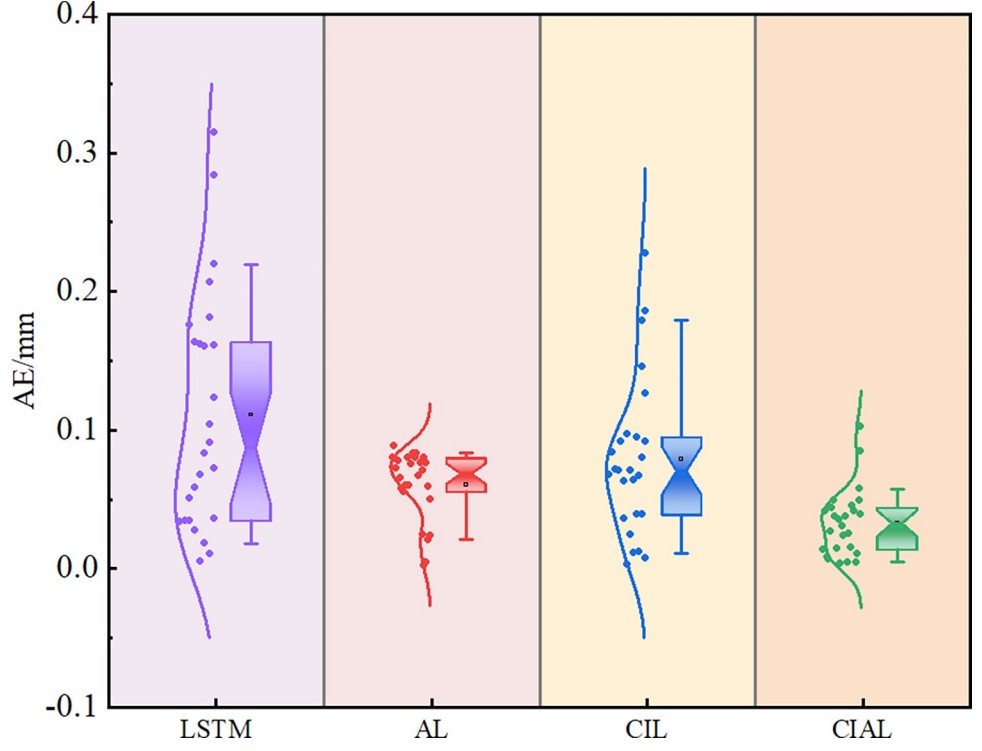

**Fig 14. Absolute error diagram.**

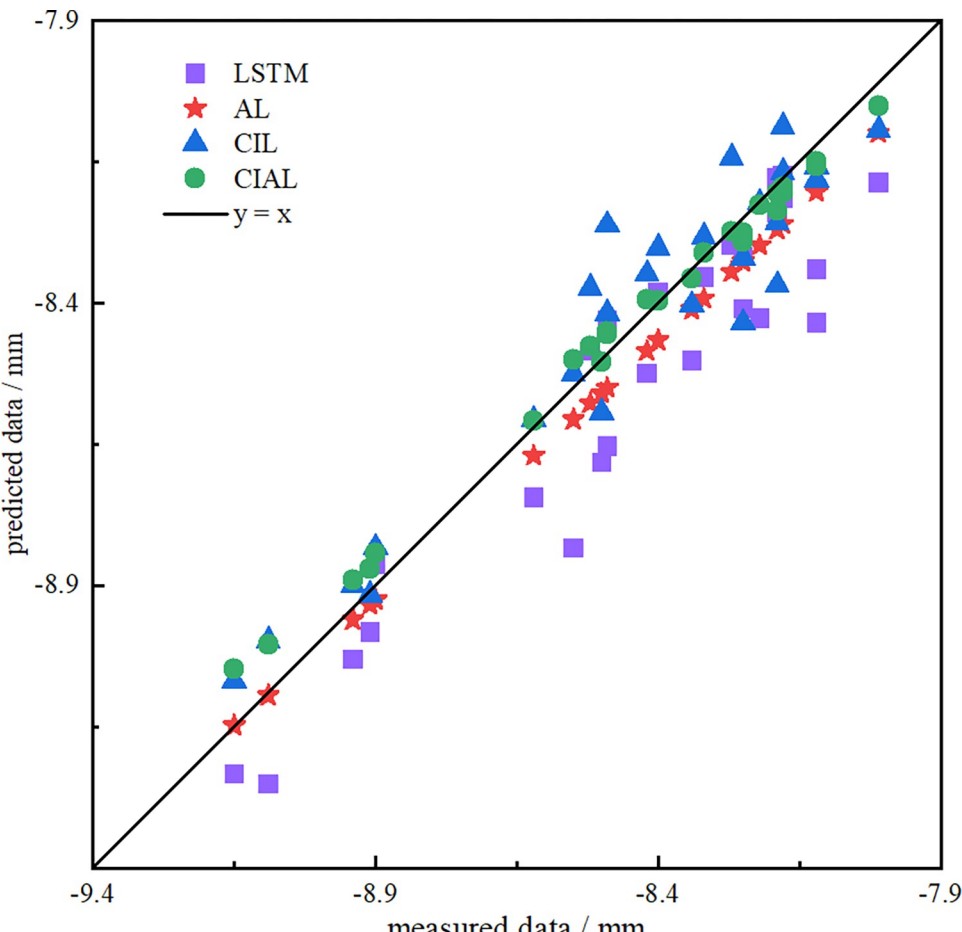

**Fig 15. Comparison of predicted and observed values of four models.**

## 5 Conclusion

This paper adopts various methods, such as complete noise-assisted aggregation empirical mode decomposition, independent component analysis, LSTM, and attention mechanism and proposes a joint noise reduction model with good effect and a dynamic prediction model with good prediction. Taking Wangjialiang Station of Urumqi Metro Line 1 as an example, different prediction models are compared and the following conclusions drawn.

First, the prediction performances of four prediction models, LSTM, AL, CIL, and CIAL, were comprehensively compared using four evaluation indexes, RMSE, MAE, $R^2$, and MAPE. The prediction accuracy on the test set was CIAL, AL, CIL, and LSTM in order, with CIAL having the best prediction performance with an RMSE of 0.041 mm, MAE of 0.033 mm, $R^2$ of 0.983, and MAPE of 0.276% and achieves the dynamic prediction of settlement data. Using this model, the construction site data can be fed back in advance, so that timely optimization and adjustment can be made in the subsequent construction process, which can provide a reference for similar projects.

Second, utilizing a CEEMDAN–ICA noise reduction model can reduce the error between the real and observed data of subway settlement deformation generated by the outside world, weaken the interference of external environment and human factors, and ensure the validity and authenticity of the data, compared with the prediction results obtained from data without

noise reduction. Accordingly, RMSE, MAE, $R^2$, and MAPE indicated improved prediction performance and improved accuracy.

Third, the addition of the attention mechanism overcomes the shortcomings of the traditional LSTM model training process, in which the information is critical and unavailable for a long time and improves the attention to the important factors associated with surface subsidence and deformation. The addition of an Adam optimization algorithm effectively improves the prediction speed. The prediction results, as indicated by RMSE, MAE, $R^2$, and MAPE revealed that omitting the attention mechanism from the data resulted in improvements in these measures, with good reliability and stability being achieved.

The CIAL model can effectively mitigate environmental disturbances and has good reliability and stability. However, the pre-processing of data for this model is still quite complex, and a simpler and more reliable model should be studied. In the future, researching more optimization algorithms to improve settlement prediction accuracy would support novel modeling and simulation techniques, and provide a safer subway settlement monitoring platform.

## Supporting information

**S1 Dataset. Original data.**
(XLSX)

**S2 Dataset. Predicted results.**
(XLSX)

**S1 Code. The code for CEEDAM.**
(PY)

**S2 Code. The code for FastICA.**
(PY)

**S3 Code. The code for AL.**
(PY)

**S4 Code. The code for LSTM.**
(PY)

**S5 Code. The code for evaluating.**
(PY)

## Acknowledgments

We are grateful to students Meng Xin and Yuanyang Chun for help, to Professors Qin Yongjun and Xie Liangfu for guidance, and to Dr. Hua for improving the language.

## Author Contributions

**Conceptualization:** Shengchao Zhu, Xin Meng.

**Data curation:** Shengchao Zhu, Xin Meng.

**Formal analysis:** Shengchao Zhu, Xin Meng.

**Funding acquisition:** Yongjun Qin.

**Methodology:** Shengchao Zhu, Yongjun Qin, Xin Meng, Yongkang Zhang.

**Resources:** Yongjun Qin, Yangchun Yuan.

**Software:** Shengchao Zhu.

**Supervision:** Yongjun Qin, Xin Meng, Liangfu Xie, Yongkang Zhang.

**Validation:** Shengchao Zhu, Yongjun Qin, Xin Meng, Liangfu Xie, Yongkang Zhang.

**Writing – original draft:** Shengchao Zhu.

**Writing – review & editing:** Shengchao Zhu.

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
