## [Decision Letter · Decision Letter 0]

21 Dec 2023

PONE-D-23-35669Prediction model of land surface settlement deformation based on improved LSTM method: CEEMDAN-ICA-AM-LSTM (CIAL) prediction modelPLOS ONE

Dear Dr. Qin,

Thank you for submitting your manuscript to PLOS ONE. After careful consideration, we feel that it has merit but does not fully meet PLOS ONE’s publication criteria as it currently stands. Therefore, we invite you to submit a revised version of the manuscript that addresses the points raised during the review process.

We look forward to receiving your revised manuscript.

Kind regards,

Dr. Muhammad Usman Tariq

Academic Editor

PLOS ONE

Journal Requirements:

"This work was supported by the Natural Science Foundation of Xinjiang Autonomous Region of China.[grant number 2021D01C073]" 

"We are grateful to the National Natural Science Foundation of China for support, to students Meng Xin and Yuanyang Chun for help, to Professors Qin Yongjun and Xie Liangfu for guidance, and to Dr. Hua for improving the language."

"This work was supported by the Natural Science Foundation of Xinjiang Autonomous Region of China.[grant number 2021D01C073]"

7. When completing the data availability statement of the submission form, you indicated that you will make your data available on acceptance. We strongly recommend all authors decide on a data sharing plan before acceptance, as the process can be lengthy and hold up publication timelines. Please note that, though access restrictions are acceptable now, your entire data will need to be made freely accessible if your manuscript is accepted for publication. This policy applies to all data except where public deposition would breach compliance with the protocol approved by your research ethics board. If you are unable to adhere to our open data policy, please kindly revise your statement to explain your reasoning and we will seek the editor's input on an exemption. Please be assured that, once you have provided your new statement, the assessment of your exemption will not hold up the peer review process.

8. PLOS requires an ORCID iD for the corresponding author in Editorial Manager on papers submitted after December 6th, 2016. Please ensure that you have an ORCID iD and that it is validated in Editorial Manager. To do this, go to ‘Update my Information’ (in the upper left-hand corner of the main menu), and click on the Fetch/Validate link next to the ORCID field. This will take you to the ORCID site and allow you to create a new iD or authenticate a pre-existing iD in Editorial Manager. Please see the following video for instructions on linking an ORCID iD to your Editorial Manager account: https://www.youtube.com/watch?v=_xcclfuvtxQ

9. Please amend either the title on the online submission form (via Edit Submission) or the title in the manuscript so that they are identical.

10. Please ensure that you refer to Figure 1, 6, 7, 8, 9, 10, 13 and 14 in your text as, if accepted, production will need this reference to link the reader to the figure.

Additional Editor Comments:

**ACADEMIC EDITOR:**

Improve the structure of the manuscriptRecheck all equationsExplain evaluating indicators and the reason to use themProvide more clarity on model training. How dataset was organised and what approach was followed. Provide the flowchartWhile conclusion is provided, also provide managerial implications, recommendations, and limitations==============================

Reviewers' comments:

Reviewer's Responses to Questions

**Comments to the Author**

1. Is the manuscript technically sound, and do the data support the conclusions?

Reviewer #1: Yes

Reviewer #2: Yes

2. Has the statistical analysis been performed appropriately and rigorously? 

Reviewer #1: Yes

Reviewer #2: Yes

3. Have the authors made all data underlying the findings in their manuscript fully available?

Reviewer #1: Yes

Reviewer #2: Yes

4. Is the manuscript presented in an intelligible fashion and written in standard English?

Reviewer #1: Yes

Reviewer #2: Yes

5. Review Comments to the Author

Reviewer #1: 1. This study is very significant and the finding is also very impressive but there are fews errors that you need to fix:

i. Error at line 55 "Moghaddasi et al.Error! Reference source not found.[10] utilized", line 137, line 304, line 309, line 314, line 386, and line 394

ii. Avoid to use number at you conclusion such as 1, 2,3. Replace it with for example "First, second, third" and so on

Reviewer #2: In Section 4.1 Performance comparison:

There is no baseline or featureless model.

For regression tasks, a baseline model always predicts the mean of the training set ignoring any input features.

In Section 3.2 Evaluating Indicator Section:

RMSE, MAPE and the other indicators mentioned there are known widely in this field.

There is no need to explain in detail and write formulas.

6. PLOS authors have the option to publish the peer review history of their article (what does this mean?). If published, this will include your full peer review and any attached files.

Reviewer #1: No

Reviewer #2: **Yes: **Daniel Agyapong

---

## [Author Response · Author response to Decision Letter 0]

23 Jan 2024

Academic editor:

1. Improve the structure of the manuscript.

Thank the editor for the valuable comments on the article. We have refined the structure of the article in the style of PLoS One.

2. Recheck all equations.

Thank you for your reminding. We've checked all the equations and made sure they're correct.

3. Explain evaluating indicators and the reason to use them.

Thank the editor for the valuable comments. We have already discussed the evaluating indicator in Section 3.2 and explained their meaning with equations.

4. Provide more clarity on model training. How dataset was organized and what approach was followed. Provide the flowchart.

Thank the editor for the valuable comments. The training process and specific methods of the model have been provided in the paper, as shown in Fig 11. (Line 353)

5. While conclusion is provided, also provide managerial implications, recommendations, and limitations.

Your suggestion really means a lot to us. The managerial implications, recommendations, and limitations have been provided after the conclusion. (Line 442~446)

Reviewer #1:

1. Error at line 55 "Moghaddasi et al.Error! Reference source not found.[10] utilized", line 137, line 304, line 309, line 314, line 386, and line 394.

Thanks for your careful checks. We have reinserted the reference [10] (Line 55) to ensure it is discovered. We have fixed the errors in line 137, line 309, line 314, line 386 and line 394. We apologize for not finding an error in line 304.

2. Avoid to use number at you conclusion such as 1,2,3. Replace it with for example "First, second, third" and so on.

Thanks for your suggestions. We have replaced the number with "First, second, third" and so on. (Line 424,431,436)

Reviewer #2:

1. In Section 4.1 Performance comparison:

There is no baseline or featureless model.

For regression tasks, a baseline model always predicts the mean of the training set ignoring any input features.

Thank the reviewer for the valuable comments. The data obtained in the experiment already possesses temporal characteristics, and since our model does not contain other features, we believe that there is no need to compare it with a featureless model.

2. In Section 3.2 Evaluating Indicator Section:

RMSE, MAPE and the other indicators mentioned there are known widely in this field.

There is no need to explain in detail and write formulas.

Thank you again for your positive comments and valuable suggestions to improve the quality of our manuscript. The description of the evaluation indicators in Section 3.2 is provided to account for the specific indicators used in this paper to assess the model, hence the need for explanation.

---

## [Editor Report · Decision Letter 1]

26 Jan 2024

Prediction model of land surface settlement deformation based on improved LSTM method: CEEMDAN-ICA-AM-LSTM (CIAL) prediction model

PONE-D-23-35669R1

Dear Dr. Qin,

We’re pleased to inform you that your manuscript has been judged scientifically suitable for publication and will be formally accepted for publication once it meets all outstanding technical requirements.

Kind regards,

Dr. Muhammad Usman Tariq

Academic Editor

PLOS ONE

---

## [Editor Report · Acceptance letter]

27 Feb 2024

PONE-D-23-35669R1 

PLOS ONE

Dear Dr. Qin, 

I'm pleased to inform you that your manuscript has been deemed suitable for publication in PLOS ONE. Congratulations! Your manuscript is now being handed over to our production team.

Kind regards, 

on behalf of

Dr. Muhammad Usman Tariq 

Academic Editor

PLOS ONE